# An Organic Fraction of *Oenothera rosea* L’Her Ex. Aiton Prevents Neuroinflammation in a Rat Ischemic Model

**DOI:** 10.3390/ph17091184

**Published:** 2024-09-09

**Authors:** Alejandro Costet-Mejía, Gabriela Trejo-Tapia, Itzel Isaura Baca-Ibarra, Aida Araceli Rodríguez-Hernández, Julio García-Hernández, Brenda Hildeliza Camacho-Díaz, Alejandro Zamilpa

**Affiliations:** 1Centro de Desarrollo de Productos Bióticos (CEPROBI), Instituto Politécnico Nacional, Yautepec 62739, Morelos, Mexico; alejandro.costet@gmail.com (A.C.-M.); bcamacho@ipn.mx (B.H.C.-D.); 2Centro de Investigación Biomédica del Sur (CIBIS), Instituto Mexicano del Seguro Social (IMSS), Xochitepec 62780, Morelos, Mexico; 3Centro Médico Nacional Siglo XXI, Instituto Mexicano del Seguro Social (IMSS), Ciudad de México 06720, Mexico; itzel.baca@imss.gob.mx (I.I.B.-I.); juliogar3@gmail.com (J.G.-H.); 4CONAHCyT-Instituto Politécnico Nacional (CEPROBI), Yautepec 62739, Morelos, Mexico; arodriguezhe@ipn.mx

**Keywords:** ischemic cascade, Onagraceae, neuronal deficit, hippocampal neurons

## Abstract

Background: *Oenothera rosea* L’Her Ex. Aiton, presenting antioxidant and anti-inflammatory activities, is traditionally used to treat bruises and headaches and as a healing agent. This study aimed to investigate whether its organic fraction (EAOr) has neuroprotective properties against neuroinflammation in the context of ischemia/reperfusion. Methods: The chemical composition of EAOr was determined using HPLC techniques, and its neuroprotective activities were evaluated in a common carotid-artery ligation model for the induction of ischemia/reperfusion (I/R). The animals were supplemented with EAOR for 15 days. On the last day, the animals were rested for one hour, following which the common carotid-artery ligation procedure was performed to induce I/R. The neurological deficit was evaluated at 24 h after I/R using Bederson’s scale, and the relative expression of inflammatory genes and structure of hippocampal neurons were analyzed at 48 h. Results: The chemical analysis revealed five major compounds in EAOr: gallic acid, rutin, ellagic acid, and glucoside and rhamnoside quercetin. EAOr prevented neurological deficit 24 h after I/R; led to the early activation of the *AIF* and *GFAP* genes; reduced *Nfkb1*, *IL-1beta*, *Il-6* and *Casp3* gene expression; and protected hippocampal neurons. Conclusions: Our findings demonstrate that EAOr contains polyphenol-type compounds, which could exert a therapeutic effect through the inhibition of neuroinflammation and neuronal death genes, thus maintaining hippocampal neurons.

## 1. Introduction

Stroke is a neurological condition caused by blood circulation disorders in the brain. It is highly prevalent and is the main cause of death and disability in the adult population [1]. Ischemic stroke (IS) is one of the most common types of stroke, accounting for more than 85% of cases; it occurs due to a sudden blockage in an artery that supplies blood to a region of the brain [2,3]. The disturbance of blood flow deprives brain tissues of oxygen, glucose, and energy (ATP), triggering dynamic changes that are part of the so-called ischemic cascade. Secondary to ischemia/reperfusion (I/R), inflammation and apoptosis are among the pathological events that occur during the ischemic cascade, which play important roles in neuronal damage [4]. Microglia and astrocytes are the first cell types to respond to ischemic injury [5]. Activated microglia produce anti- and pro-inflammatory cytokines to promote damage or facilitate repair, depending on the activation signals. Meanwhile, astrocytes can exacerbate ischemic lesions or limit their expansion through neurotrophic factors [6,7]. Therefore, neuroinflammation is an essential process that defines functional prognosis. However, the complexity of the neuroinflammatory process in different stages of ischemic stroke makes it difficult to find a specific target for the attenuation of neuroinflammation [8].

At present, medical therapy in this context remains limited and involves the use of recombinant tissue plasminogen activator (tPA), a thrombolytic agent, to dissolve clots. However, the therapeutic window is limited and is only applied in less than 10% of patients, due to the risk of adverse effects [9,10]. Additionally, reperfusion after thrombolysis can aggravate the lesion and reduce the benefits of therapeutic action. Alternative drugs, such as naturally occurring compounds, have attracted increasing attention as they have the potential to limit neuroinflammation and support neuronal survival. *Oenothera rosea* L’Her Ex. Aiton (Onagraceae), known traditionally as “*hierba del golpe*”, is traditionally used in Mexico to relieve headaches and bruises as well as to reduce inflammation and heal wounds [11,12,13]. Additionally, this species displays anti-inflammatory properties and reduces the expression of cytokines such as IL-1beta, Il-6, and TNF-α. It also reduced the formation of free radicals in models of chronic and acute inflammation [14,15,16]. These therapeutic properties have been attributed to its chemical composition, which is dominated by phenolic compounds such as gallic acid, apigenin, rutin, and kaempferol [15,17]. It has been reported that polyphenols may exert neuroprotective effects in models of ischemic stroke [18].

In this study, we hypothesized that treatment with an ethyl acetate fraction of *O. rosea* (EAOr) plays an important role in ischemia/reperfusion (I/R) neuroinflammation by acting on the expression of gene markers of the ischemic cascade. We expected this effectiveness to be related to the set of chemical compounds present in the EAOr.

## 2. Results

### 2.1. Chemical Composition of EAOr

Chemical separation of the ethyl acetate fraction of *O. rosea* (EAOr) via bipartition was confirmed based on retention time and UV spectrum data. According to HPLC analysis (Figure 1), the EAOr consisted mainly of phenolic compounds with eluting times between 8 and 11 min. Chemical fractionation allowed us to identify five compounds: rutin (1), quercetin 3-D-glucoside (2), gallic acid (3), ellagic acid (4), and quercetin rhamnoside (5) (Figure 2). The isolated compounds were identified based on detailed spectroscopic analyses, compared to commercial standards (Appendix A). Therefore, *O. rosea* contains several polar compounds that can exert the expected therapeutic effects.

### 2.2. EAOr Attenuates Global Neurological Deficit

To determine whether I/R was successful, 24 h after recovery from anesthesia, all rats were evaluated using the score described by Bederson [19]. After 1 day of survival, the stroke group displayed significant (*p* < 0.05) neurological deficits, including retraction of the front legs and loss of grip strength (1–3 score), when compared to the control group (0 score). The experimental rats that received telmisartan displayed retraction of the front legs (1 score). The EAOr-treatment group had a better neurological score than the stroke group, maintaining mostly non-deficit conditions (0–1 score) (Figure 3). These results indicated that the ischemic stroke model was established successfully and that the EAOr treatment mitigated the deficit after I/R.

### 2.3. EAOr Induces Gene Expression of AIF-1 and GFAP Related to Microglia and Astrocyte Pathways

Ischemic stroke is triggered by a complex series of biochemical processes, originating from the ischemic cascade and finally inducing cell death. To compare the relative gene expression of all groups at 2 days after ischemia/reperfusion injury, an analysis of inflammatory marker genes (*AIF-1*, *GFAP*, *NF-κB*, IL-6, and *IL-1β*) was performed using RT-qPCR assays.

First, *AIF-1* and *GFAP* mRNA expression levels were assessed to detect the activation of microglia and astrocytes, respectively. The results showed that *AIF-1* gene expression was similar between the stroke group (0.018-fold) and the control group (0.0192-fold). Meanwhile, the transcription level of this gene increased significantly in the telmisartan (0.0278-fold) and EAOr (0.02314-fold) groups (Figure 4A). Figure 4B shows the expression of the *GFAP* gene in all groups with I/R. Telmisartan had similar expression levels of *GFAP* (0.0007-fold) to the stroke group (0.0008-fold). In the EAOr group, *GFAP* expression increased significantly, by more than threefold (0.0028-fold), when compared to the stroke group. These results indicate that EAOr could induce the early activation of microglia and astrocytes after I/R.

### 2.4. Expression of Inflammatory Cytokine Genes IL-1beta and Il-6 Was Repressed by EAOr through Nfkb1

Activated microglia and astrocytes produce additional cytokines that can lead to brain tissue damage. We quantified the expression levels of *Nfkb1*, *Il-6*, and *IL-1beta* genes as markers of inflammation at 2 days after I/R. In the stroke group, the *Nfkb1* gene (0.1601-fold) was strongly induced, compared to the control group (0.0066-fold). Additionally, in the telmisartan group, mRNA expression was higher (0.0448-fold) than in the control group but twofold lower than in the stroke group. In contrast, the EAOr fraction attenuated *Nfkb1*gene expression (0.0030-fold), compared to the stroke group (Figure 5A).

Regarding the expression levels of the *Il-6* and *IL-1beta* genes, in the stroke group, there was no difference in the expression of the *Il-6* gene (0.00013-fold) as compared to the control group (0.00011-fold); meanwhile, there was a significant increase in the expression of this gene in the telmisartan group (0.00019-fold) relative to the control. To the contrary, in the EAOr group, the *Il-6* gene expression was twofold lower (4.7 × 10^5^-fold) than in the stroke group (Figure 5B).

Next, the expression level of the *IL-1beta* gene was determined in the ischemia/reperfusion groups. The stroke group (6.47 × 10^5^-fold) was the only group that differed significantly from the control group (2.437 × 10^5^-fold). The telmisartan (4.647 × 10^5^-fold) and EAOr (5.747 × 10^5^-fold) groups presented lower expression than the stroke group (Figure 5C). Therefore, EAOr treatment could inhibit inflammatory markers through *Nfkb* gene expression after I/R.

### 2.5. Casp3 Gene Expression Decreased with Administration of EAOr

Caspase 3 (Casp3) is considered a mediator of apoptosis and is implicated in I/R. We observed an increase in the expression of the *Casp3* gene in the stroke (0.0062-fold) and telmisartan (0.0121-fold) groups, which was more significant in the latter group than the control group (0.0041-fold), while the EAOr fraction significantly decreased the expression of *Casp3* (0.0034-fold), when compared to the stroke group (Figure 6). This result indicates that the administration of EAOr in the context of ischemic stroke could restore *Casp3* expression in the cerebral tissue of rats.

### 2.6. Neuroprotection of the Hippocampal Area by EAOr

To determine the effect of EAOr on neuronal degeneration in the hippocampus after I/R, we performed histopathological analysis with blue toluidine staining. As shown in Figure 7A, the control group had clear membranes and ordinarily shaped and centered nuclei in the hippocampal CA3 area. In contrast, the stroke group showed pyramidal neurons, and the nuclei were shrunken, condensed and, in some cases, disintegrated (Figure 7B). The telmisartan group was similar to the stroke group (Figure 7C). These changes were less severe in the EAOR group, and the neurons presented a better appearance (Figure 7D). Therefore, in the EAOr group, the brain tissue structure was better conserved after ischemic stroke.

## 3. Discussion

The results suggested that *O. rosea* had several beneficial properties related to inflammation in the I/R-induced rats. Phytochemical analyses indicated that the EAOr fraction contains five main phenolic compounds: gallic acid, rutin, ellagic acid, and glucoside and rhamnoside of quercetin. Based on the results, we propose that EAOr improves neurobehavioral capacity through activating microglia- and astrocyte-related genes (*AIF-1* and *GFAP*, respectively), as well as by mitigating the expression of gene markers of inflammation (*Nfkb1*, *Il-6*, and *IL-1beta)* and apoptosis (*Casp3*). Additionally, EAOr maintained the neuronal protection of the hippocampus.

The common carotid-artery occlusion model was introduced to study stroke-like cerebral hypoperfusion and evaluate the effects of medications [20]. I/R leads to changes in biochemical mechanisms (ischemic cascade), which worsen over time and culminate in neuronal death. The first signs of I/R injury occur in the cortical and motor cognitive areas, which are affected by the lack of blood flow to the brain [21]. In the present study, the administration of EAOr treatment reduced the degree of neural deficit 24 h after I/R induction. It is known that natural products such as polyphenols have several health benefits, notably including neuroprotective effects. Gallic acid has been reported to improve passive memory in the rat I/R model, due to its antioxidant activity mediated through the regulation of mitochondrial dysfunction [22,23]. Similar effects have been reported for kaempferol, quercetin, and rutin, which reduced the formation of malondialdehyde (MDA) and increased the levels of the antioxidant enzymes superoxide dismutase (SOD) and catalase (CAT) after I/R [24,25].

The interruption of the supply of glucose and oxygen leads to a series of neurochemical mechanisms that evolve over time and space, culminating in cell death [26,27]. In the EAOr group, we observed an upregulation of the microglia (*AIF-1*) and astrocyte (*GFAP*) genes 48 h after I/R. Simultaneously, EAOr reduced the expression of inflammatory cytokine genes, inhibited the expression of *Nfkb1* and *Il-6,* and maintained a basal level of the *IL-1beta* gene.

Reports on anti-inflammatory signaling have indicated that polyphenols promote the M2 polarization of microglia and reduce neuroinflammation [28]. Several studies have proposed that polyphenols such as gallic acid, rutin, quercetin, and kaempferol promote the M2 polarization of microglia through the modulation of PI3K/Akt/NF-κB signaling. Therefore, the inhibition of the expression of pro-inflammatory proteins could reduce the degree of edema in the early stage after I/R and lead to improved neurological function [29,30,31,32,33,34].

The roles of mitochondria in programmed cell death or apoptosis are triggered by integrated signals from several sources such as the activity of NF-κB induced by TNFα. Reports have indicated that NF-κB leads to the release of cytochrome c, which is responsible for the activation of caspases [35].

Our data suggest that the effect of EAOr on the *Casp3* gene was dependent on the inhibition of *Nfkb* and inflammatory cytokines genes, as EAOr inhibited the expression of the *Casp3* gene at 48 h after I/R. We speculate that this effect could contribute to the therapeutic capacity of the chemical compounds present in EAOr.

Finally, histological analysis confirmed that in the stroke group, I/R produced morphological changes in the hippocampal neurons of the CA3 region, while these same neurons in the animals that received EAOr were more resistant to hypoxia/reoxygenation. In this study, we identified important effects of *O. rosea* on inflammatory mechanisms occurring after I/R. We hypothesize that the inhibition of genetic markers involved in the ischemic cascade could safeguard the integrity of neurons in the face of hypoxia/reoxygenation.

## 4. Materials and Methods

### 4.1. Plant Material

*Oenothera rosea* L’Her Ex. Aiton was cultivated in Xochitepec, Morelos, Mexico (18°47′02″ N, 99°13′49″ W). It was identified by Dr. Arturo Mora-Olivo of the Applied Ecology Institute at the Autonomous University of Tamaulipas and assigned the voucher number 03358 [16].

### 4.2. EAOr Fraction Preparation

The aerial parts of *O. rosea* were collected in the months of December–July, then dried at room temperature and under shade for four days and milled to obtain 4–6 mm. The material (600 g) was extracted by maceration with a 70% ethanol and 30% water solution for 48 h at 24–27 °C. Subsequently, the extract was filtered (Whatman No. 4 paper) and concentrated using a rotary evaporator at 50 °C under reduced pressure, in order to obtain the hydroalcoholic extract (HAOr). This extract was re-suspended in 400 mL of a 50:50 water–ethyl acetate solution and concentrated using the rotatory evaporator (at 42 °C, 0.2–0.3 bar) to obtain the organic fraction (EAOr). EAOr was lyophilized and tested in the biological model used for this study.

### 4.3. Chemical Separation

The EAOr fraction (8.0 g) was further fractioned using open-column chromatography. Silica gel (600.063–0.200, Merck) was used as the stationary phase, and a dichloromethane–methanol gradient system (100:0, 98:2, 96:4, 94:6, 92:8, 90:10, 88:12, 86:14, 84:16, 82:18, 80:20, 70:30, 50:50, and 0:100) was used as the mobile phase. Normal-phase thin-layer chromatography (Silica gel 60 F254, Merck, MA, USA) was used for the analysis of samples (50 mL) for the chemical separation of EAOr, with compounds grouped by chemical similarity.

### 4.4. Isolation and Characterization of Compounds of O. rosea

Chemical characterization was performed via HPLC analysis using a Waters 2696 separation module system equipped with Empower Pro Software (Version 3, Waters Corporation, Milford, MA, USA). The column used was a Supelcosil analytical LC-F (4.6 mm × 250 mm, 5 μm particle size; Sigma-Aldrich, Bellefonte, PA, USA). The gradient system consisted of acidulated water (solvent A) and acetonitrile (solvent B), with the following gradient system: 0–1 min, 0%; 2–3 min, 5%; 4–20 min, 30%; 21–23 min, 50%; 24–25 min, 80%; 26–27 min, 100%; and 28–30 min, 0%. The flow rate was maintained at 0.9 mL/min, and the sample injection volume was 10 µL [36]. The chromatograms were visualized at 280 nm for gallic acid, 255 nm for ellagic acid, and 350 nm for the flavonoids. Five ascendant concentrations of each commercial standard were used to build the calibration curves as follows: gallic acid (12.5, 25, 50, 100, and 200 µg/mL; Y = 11,949X + 30,778; R^2^ = 0.999), ellagic acid (1.56, 3.185, 6.25, 12.5 and 25 mg/mL; Y = 88,507X + 46,190; R^2^ = 0.999), rutin (12.5, 25, 50, 100, and 200 µg/mL; Y = 24,399X − 6087; R^2^ = 0.999), quercetin glucoside (12.5, 25, 50, 100, and 200 µg/mL; Y = 17,308X + 17,899; R^2^ = 0.999), and quercetin rhamnoside (12.5, 25, 50, 100, and 200 µg/mL; Y = 4789.5X − 26857; R^2^ = 0.994). The EAOr treatment contains 10.6 mg of gallic acid, 5.7 mg of ellagic acid, 1.25 mg of rutin, 1.17 mg of quercetin glucoside, and 15.1 mg of quercetin rhamnoside per gram of this organic fraction.

### 4.5. Animals

Experiments were performed on male Sprague–Dawley rats (SD) (300–350 g), obtained from the Bioterium of the Center of Health Research Coordination of the Century XXI, Medical Center (Mexico City). The protocol was approved by the local Ethics Committee (R-2022-1701-010) and adhered to the Mexican Official Norms (NOM-062-ZOO-1999). Every effort was made to reduce the number of animals used and to ensure that their suffering was kept to a minimum.

Animals were housed in 10 rats per group and acclimatized for 7 days at 22–24 °C, with a 12-h light/dark cycle and food and water provided ad libitum. The groups were assigned as follows: control, stroke, telmisartan (40 mg/kg, v.o), and EAOr (25 mg/kg, v.o). In the telmisartan and EAOr groups, the corresponding treatment was administered orally by cannula, while the control and stroke groups received water in the same way, once per day for 15 days. On the 15th day, the carotid-artery occlusion surgery was performed 60 min after the last administration.

The safety of the ethyl–acetate fraction of the aerial parts of *O. rosea* (EAOr) was examined according to the OECD guidelines, using 16 female mice divided into four groups. The first three groups were given increased oral doses, ranging from 100 to 1000 mg/kg b.wt. of the treatment. The last group received distilled water. Animals were observed periodically during the first 24 h for 14 days. With these results, the considered dose was 25 mg/kg.

Based on previous work, telmisartan has been shown to reduce markers of neuroinflammation such as glial fibrillary acidic protein (GFAP) and metalloproteinase-9 (MMP-9), as well as to be a PPAR agonist. Therefore, telmisartan was used as a pharmacological control at a dose of 40 mg [37,38,39].

### 4.6. Ischemia Induction by Carotid-Artery Occlusion

The cerebral ischemia-reperfusion model was performed using the common carotid-artery occlusion method. The animals were anesthetized with ketamine (10%) and xylazine (2%) before the surgery. Subsequently, the neck was shaved and disinfected to make the incision in the midline of the neck, carefully exposing the left and right common carotid arteries, which were ligated with clamps. After 75 min of occlusion, the clamps were removed to restore blood flow, and the skin was sutured. In the control group, the same procedure was followed, except that the common carotid arteries were not occluded. The animals’ body temperature was kept at 26–28 °C during the surgery using a heating pad.

### 4.7. Evaluation of Neurological Function

The global neurological analysis was performed at 24 h after the surgical procedure. To ensure accuracy, the test was conducted by experienced observers who were blinded to the groups. Bederson’s scale was used to examine the rats’ neuronal deficits, following a previously published protocol [19]. Parameters, such as forelimb flexion, resistance to lateral thrusting, and circular behavior were evaluated. Functions were considered normal when the score was zero, a moderate deficit with one point and a severe deficit with two or three points. One point was awarded for each test not completed or for the presence of signs of injury.

### 4.8. RNA Extraction

Animals were sacrificed at 48 h after surgery for I/R via transcardial perfusion followed by fixation with 4% paraformaldehyde. The brains were removed and kept in a cold chain (−60 to −80 °C) to prevent degradation until they were used for RNA extraction. Total RNA extraction was performed using TRIzol reagent (Invitrogen, Carlsbad, CA, USA) as recommended by the manufacturer. The brain was macerated and immersed in 1 mL of TRIzol, followed by 0.2 mL of chloroform. For isolation, 0.5 mL of isopropanol was added to the aqueous phase containing the RNA. Then, the pellet was washed in 75% of ethanol–DEPC and then resuspended in 20–50 μL of H_2_O–DEPC.

The integrity of RNA was checked through 1.2% agarose gel electrophoresis under denaturing conditions and spectrophotometry (Nanodrop 2000, Thermo Scientific, Wilmington, DE, USA) with a 260/280 nm absorbance ratio.

### 4.9. Reverse Transcription-Quantitative Polymerase Chain Reaction (RT-qPCR)

Quantitative PCR was implemented on StepOne Real-Time PCR Detection System using the StepOne Software v2.1 (Applied Biosystems, Foster City, CA, USA) and iTaq Universal SYBR Green II One-Step kits. All samples were amplified in triplicate with the following protocol: 10 min at 50 °C (cDNA synthesis), 1 min at 95 °C (polymerase activation), followed by 40 cycles at 95 °C (DNA denaturation), and 1 min at 60 °C for *β-actin*, *caspase-3,* and *AIF* genes; 61 °C for *GFAP* gene; or 55 °C for *IL-6* and *IL-1β* genes. The melting curve was generated with cycles of 5–95 °C for 15 s, increasing the temperature by 0.5 °C each cycle. We used the *β-actin* gene as a reference in all experiments, and the mRNA levels of candidate genes were determined using the 2^−ΔΔCT^ method [40]. Each sample was analyzed in triplicate (*n* = 3) (Table 1).

### 4.10. Histopathology Staining

Two days after ischemia/reperfusion, the rats from each group were anesthetized with 1% pentobarbital sodium (100 mg/kg, intraperitoneally). The brains were removed and fixed in 4% paraformaldehyde and then embedded in paraffin following standard methods [41]. The paraffin-embedded blocks were cut into a series of 7 µm thick slices of sections coronally using a microtome (Leica RM 2125 RT, Dallas, TX, USA) and stained with 1% toluidine blue (Sigma Aldrich, Saint Louis, MO, USA). The morphology of neurons in the CA3 hippocampal area were observed (at ×400 magnification) under a light microscope (Nikon Eclipse 80i, Tokyo, Japan).

### 4.11. Statistical Analysis

The data are expressed as the means ± standard error of the mean (S.E.M). One-way analysis of variance (ANOVA) followed by a Bonferroni test was used to compare relative gene expression. Differences were considered significant when *p* < 0.05. The software used was GraphPad Prism Version 9 (GraphPad Software, San Diego, CA, USA).

## 5. Conclusions

The present study demonstrated the neuroprotective effect of EAOr against acute I/R in SD rats. In particular, the administration of EAOr prevented motor dysfunction in the rats, regulated the early response of microglia and astrocytes, and downregulated the inflammatory response mediated by the *IL-1beta* and *Il-6* genes. Furthermore, we propose an anti-apoptotic effect of the EAOr, mediated by the inhibition of *Casp3* gene expression. These findings provide new pharmacological evidence of the potential of *Oenothera rosea* in the therapeutic treatment of ischemia/reperfusion injury.

## Figures and Tables

**Figure 1 pharmaceuticals-17-01184-f001:**
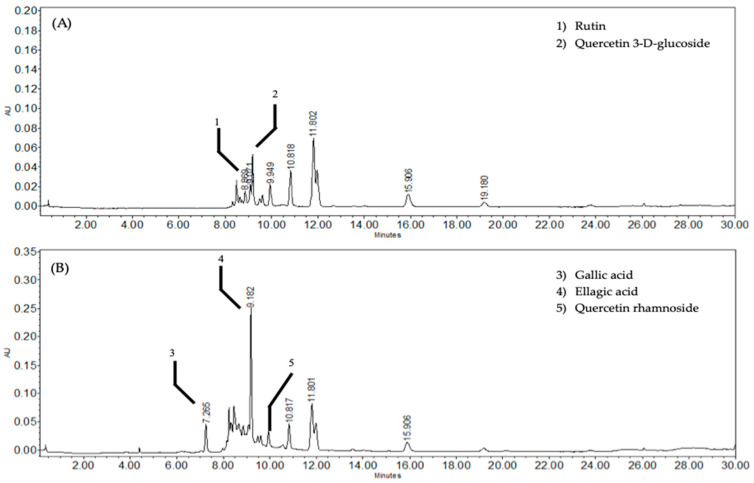
HPLC chromatograms (detection at 360 nm); (**A**) ethyl-acetate fraction (EAOr), with rutin at 8.86 min, and quercetin-3-D-glucoside at 9.07 min; (**B**) EAOr (280 nm), with gallic acid at 7.26 min, ellagic acid at 9.18 min, and quercetin rhamnoside at 9.94 min.

**Figure 2 pharmaceuticals-17-01184-f002:**
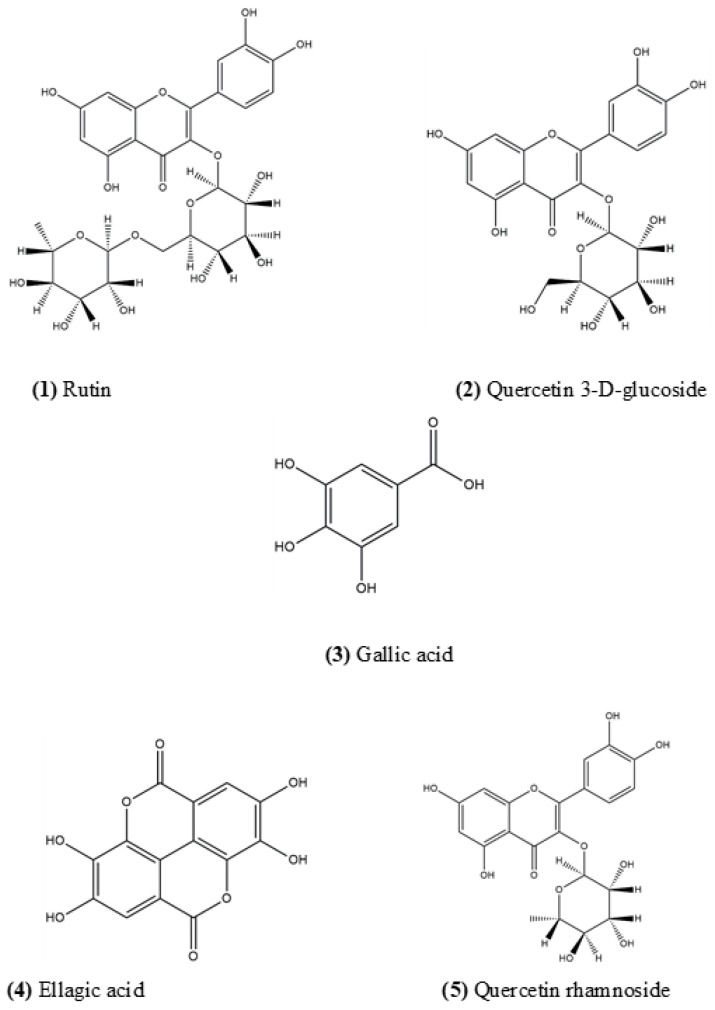
Chemical structure of compounds characterized in EAOr fraction. (**1**) rutin, (**2**) quercetin 3-D-glucoside, (**3**) gallic acid, (**4**) ellagic acid, and (**5**) quercetin rhamnoside.

**Figure 3 pharmaceuticals-17-01184-f003:**
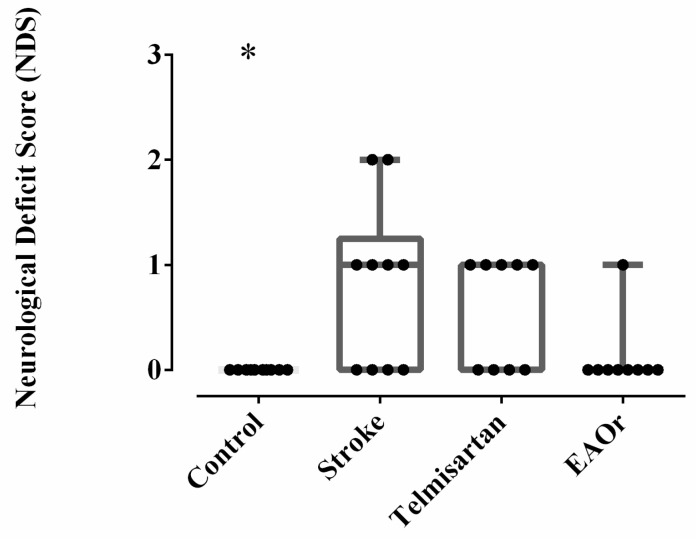
Mice were subjected to 75 min of I/R, and neurological and motor function were assessed 24 h later. *n* = 9–10 per group. Groups were compared through ordinary one-way ANOVA with a Bonferroni adjustment test. (*) *p* < 0.05; vs. Stroke group. The dots refer to each animal.

**Figure 4 pharmaceuticals-17-01184-f004:**
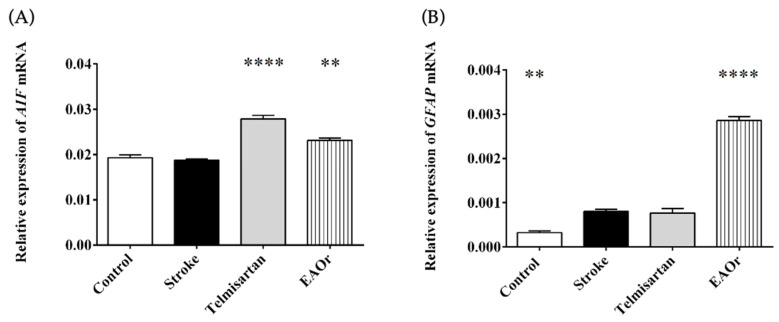
Relative expression of genes associated with microglia (AIF, (**A**)) and astrocytes (GFAP, (**B**)) after ischemia/reperfusion under each of the treatments: telmisartan, and EAOr. Groups were compared through ordinary one-way ANOVA with a Bonferroni adjustment test. (**) *p* < 0.01, and (****) *p* < 0.0001; vs. stroke group.

**Figure 5 pharmaceuticals-17-01184-f005:**
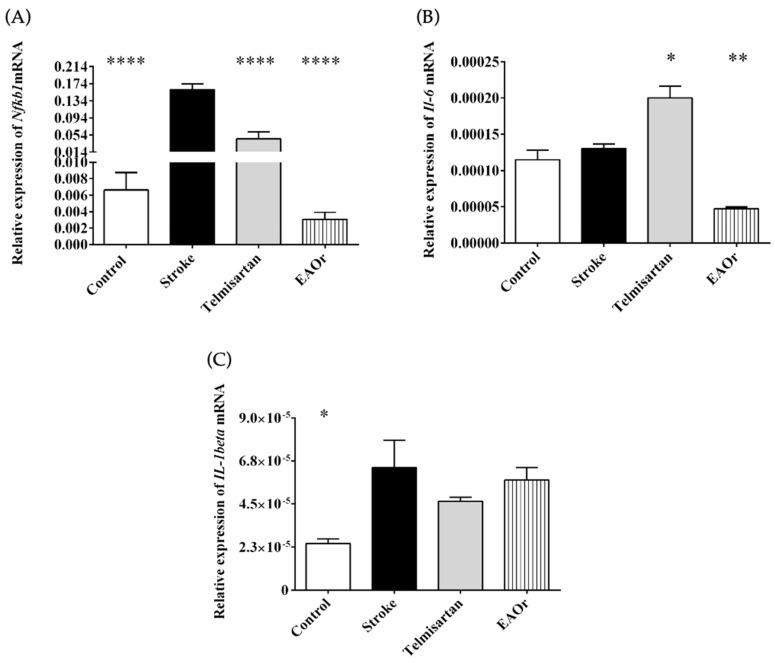
Relative expression of genes associated with inflammation: (**A**) *Nfkb1*, (**B**) *Il-6*, and (**C**) *IL-1beta*, after ischemia-reperfusion under each of the treatments: telmisartan; and EAOr. Groups were compared using ordinary one-way ANOVA with a Bonferroni adjustment test. (*) *p* < 0.05, (**) *p* < 0.01, and (****) *p* < 0.0001; vs. stroke group.

**Figure 6 pharmaceuticals-17-01184-f006:**
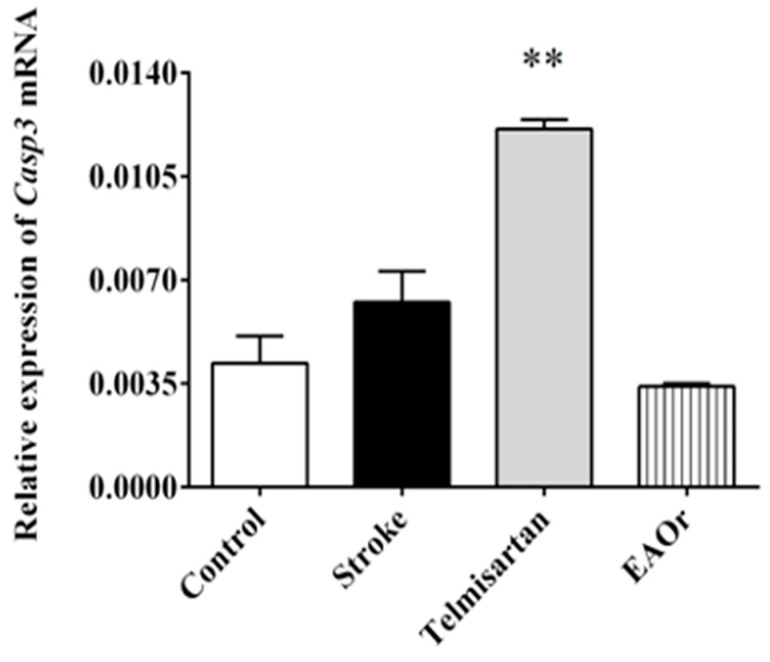
Relative expression of *Casp3* gene associated with inflammation, after ischemia/reperfusion under each of the treatments: telmisartan; and EAOr. Groups were compared using ordinary one-way ANOVA with a Bonferroni adjustment test. (**) *p* < 0.01; vs. stroke group.

**Figure 7 pharmaceuticals-17-01184-f007:**
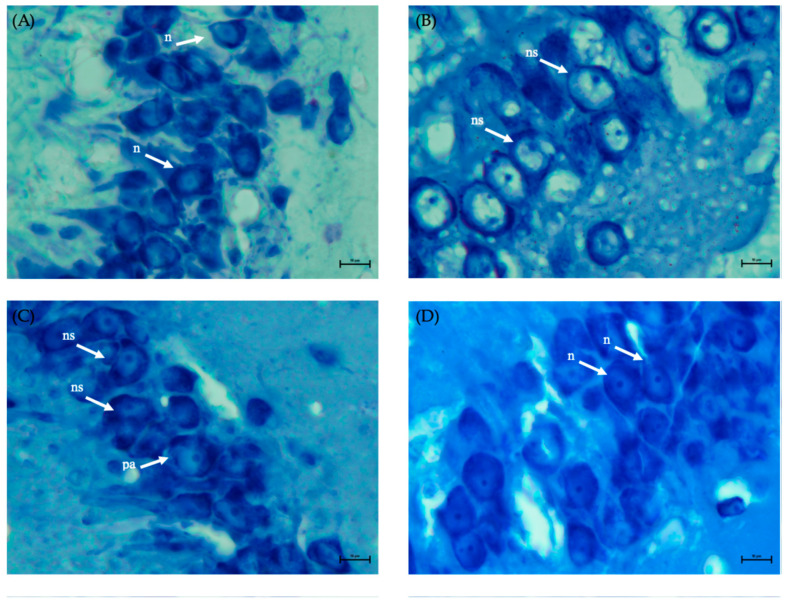
Photomicrographs of the CA3 region of brains of ischemia/reperfusion model animals: (**A**) Control group, (**B**) Stroke group, (**C**) Telmisartan group, and (**D**) EAOr group. Brain tissue was stained with toluidine blue. The morphology of neurons in the CA3 hippocampal area were observed under a light microscope (×400 magnification). The bar in the lower right corner represents 10 µm. n: normal neuron, ns: neurons shrunken, pa: possible apoptosis.

**Table 1 pharmaceuticals-17-01184-t001:** Primer sequences and their amplification temperatures for each gene.

Gene	Sequences (5′–3′)	Amplification Temperature
*Casp3*	F: GGAGCTTGGAACGCGAAGAA R: ACACAAGCCCATTTCAGGGT	60 °C
*AIF-1*	F: TCTGAATGGCAATGGAGATA R: GTTGGCTTCTGGTGTTCT	60 °C
*GFAP*	F: TTGGCTTATGTTCTGTCCATTGAG R: AGAGTGGTATCGGTCCAAGTT	61 °C
*IL-1beta*	F: GGGTCTGACTCCCATTTTCC R: TCTGTGACTCGTGGGATGATGAC	55 °C
*Il-6*	F: CAGAGTCATTCAGAGCAATAC R: CTTTCAAGATGAGTTGGATGG	55 °C
*Nfkb1*	F: TTCCCTGAAGTGGAGCTAGGA R: CATGTCGAGGAGACACTGGA	60 °C

## Data Availability

Data are contained within the article.

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
