# Peer review of "An Organic Fraction of Oenothera rosea L’Her Ex. Aiton Prevents Neuroinflammation in a Rat Ischemic Model"

_pharmaceuticals, 2024, doi:10.3390/ph17091184_

Round 1

Reviewer 1 Report

Comments and Suggestions for Authors

I want to express my appreciation for the opportunity to review this manuscript. I find the work to be quite interesting, even though it would have been beneficial to include additional experimental investigations to provide a more comprehensive understanding of the study. I just have a few minor queries that I would like to clarify.

Results: line 69: The "R" in "EAOR" should be lowercase

Figure 2: In the caption, it would be better to indicate the names of the molecules corresponding to the numbers.

Result 2.1.5.: Even though the text consistently refers to expression levels, it would be interesting to evaluate how caspase-3 activation levels vary under different conditions through Western blot analysis, focusing on the cleaved form.

Figure 7: The images do not appear to be of high quality (especially the scale bar, which is very blurred)

Graphical abstract: line 156: What does "The text continues here" refer to? Moreover, instead of the charts, the graphical abstract could be replaced with something more representative. Alternatively, in the HPLC graph, the names of the identified polyphenolic compounds could be indicated, and the names of the analyzed transcripts and how they vary depending on the treatment (with arrows or something similar) could be added in place of the histogram.

Materials and Methods: line 226-227: In the text, it seems that a part is missing, as the sentence ends incomplete. Additionally, the functionality of telmisartan is not explicitly explained in the text.

Reviewer 2 Report

Comments and Suggestions for Authors

The manuscript requires deep English language improvement. There are many grammar and structure flaws within the whole manuscript which does not match with the journal criteria. 

Abstract section: 
Please revise the author guidlines for writing the abstract. 

Line 13, Paraphrase this sentence or include it with the previous one. 

Line 17, mention briefly those techniques, ex, HPLC or LC-MS/MS. 

Line 18, please paraphrase this sententence, rats supplemented with EAOr  (duration of supplementation). 

Results Section: 

The authors did not mention the results obtained from hydroalcoholic extract (Figure 1A). The supplementary file do not contain the standards of Figure 1A result. I am wondering the reason of analyzing hydroalcoholic extract with HPLC while the treatment is EAOr. 

The UV wavelength was not mentioned in the methods. 

To support the current findings, the authors must perform total phenolic content and total flavonoid content for the fraction used in the treatment. 

Line 63, Please mention exactly the literature or database used to compare the results.

Line 67, Please mention those polyphenols name.

Line 73, Please indicate each number to its scientific name.

Line 84, Figure 3 should be restructured again. It is ambiguous. 

Materials and MEthods section:

line 194, Please add the season and the estimated time of collection. 

Line 199, Please explain the drying process. 

line 200, Please explain how the filtration process has been occured. what kind of filtration paper did the author use? 

line 201, Please mention the pressure in numbers. 

Line 208, This procedure require further explanation. 60 samples of which extract? 

Line 223, please explain how the supplementation was provided? forced-feed? or injection?. Also, you need to mention the timing and technique used to supplement the rats with extract. Also, the authors should justify the dosage used in both treatments. Based on what, the authors have decided the dosage?.

Line 226, please revise the sentence. 

Line 246, After removing the brain, how the authors preserve the tissue? Which liquid or technique was used to keep the tissue for RNA extraction. Also, for how long dose the tissue presevered? 

Line 273- 277, please revise thses sentences. it is not proper to be placed under this section.

Comments on the Quality of English Language

The English language and the writing style should be revised and constructed cohesively. 

Round 2

Reviewer 2 Report

Comments and Suggestions for Authors

The authors have declared all the comments and the manuscript was edited accordingly.